# See, Act, Adapt: Active Perception for Unsupervised Cross-Domain Visual Adaptation via Personalized VLM-Guided Agent

Tianci Tang [*1]  Tielong Cai [*1]  Hongwei Wang [1]  Gaoang Wang [†1]

## Abstract

Pre-trained perception models excel in generic image domains but degrade significantly in novel environments like indoor scenes. The conventional remedy is fine-tuning on downstream data which incurs catastrophic forgetting of prior knowledge and demands costly, scene-specific annotations. We propose a paradigm shift through Sea$^2$ (**Se**e, **A**ct, **A**dapt): rather than adapting the perception modules themselves, we adapt how they are deployed through an intelligent pose-control agent. Sea$^2$ keeps all perception modules frozen, requiring no downstream labels during training, and uses only scalar perceptual feedback to navigate the agent toward informative viewpoints. Specially, we transform a vision-language model (VLM) into a low-level pose controller through a two-stage training pipeline: first fine-tuning it on rule-based exploration trajectories that systematically probe indoor scenes, and then refining the policy via unsupervised reinforcement learning that constructs rewards from the perception module's outputs and confidence. Unlike prior active perception methods that couple exploration with specific models or collect data for retraining them, Sea$^2$ directly leverages off-the-shelf perception models for various tasks without the need for retraining. We conducted experiments on three visual perception tasks, including visual grounding, segmentation and 3D box estimation, with performance improvements of 13.54%, 15.92% and 27.68% respectively on dataset ReplicaCAD.

## 1. Introduction

Large-scale visual models pre-trained on internet-scale imagery have demonstrated remarkable generalization across

---
*Equal contribution  [1]Zhejiang University, China. Correspondence to: Gaoang Wang <gaoangwang@intl.zju.edu.cn>.

*Proceedings of the $43^{rd}$ International Conference on Machine Learning*, Seoul, South Korea. PMLR 306, 2026.

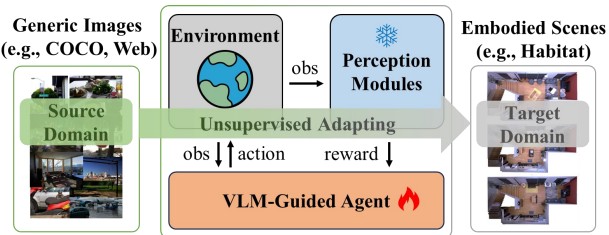

*Figure 1.* **Overview of unsupervised cross-domain adaptation via VLM-guided active perception agent.** (Left) Source domain including generic images (e.g., COCO (Lin et al., 2015), Web) where perception models pre-trained on. (Middle) Our embodied agent actively explores the indoor environment and adjusts its camera pose to capture information-rich observations that maximize perception quality. (Right) Target domain including embodied scenes (e.g., Habitat (Savva et al., 2019)) where our VLM-guided agent trained on to unsupervised adapt the domain gap.

recognition, segmentation, and visual grounding tasks (Xiao et al., 2024; 2023; Ravi et al., 2024; Ren et al., 2024; He et al., 2021; Cheng et al., 2024). However, when deployed in novel embodied environments such as indoor scenes, their performance degrades sharply due to domain gaps in viewpoint distribution, occlusion patterns, and spatial semantics (Yang et al., 2019; Ammirato et al., 2017; Zeng et al., 2020; Fan et al., 2024). The prevailing remedy is fine-tuning perception modules on downstream data, while it incurs two critical limitations: catastrophic forgetting of prior knowledge and the prohibitive cost of acquiring scene-specific annotations (e.g., pixel masks, 3D bounding boxes, or referring expressions). This raises a fundamental question: *Can we adapt perception to new domains without touching the models themselves?*

We propose a paradigm shift: instead of adapting perception modules themselves, we adapt how they are deployed. Our key insight is that perceptual performance depends not only on model capacity but also critically on the informativeness of the observation viewpoint (Yang et al., 2019; Fan et al., 2024). By intelligently controlling the agent's pose to seek out informative views, we can recover performance drops without retraining, annotating data, or modifying the perception system. Crucially, our approach freezes all perception modules, requiring no downstream labels during agent training; it relies solely on scalar feedback derived directly from

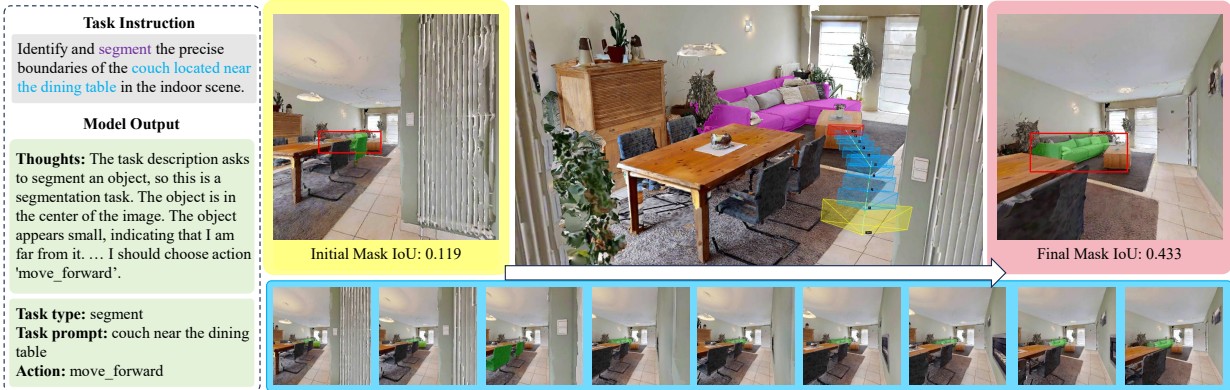

*Figure 2.* **Illustration of the active perception process for segmentation.** The agent decomposes the instruction into task-specific metadata and reasons about the scene to execute camera-adjusting actions. Starting from a highly occluded initial view (yellow) where the perception output (green mask) for the target object (red box) is poor, it follows a trajectory of navigational steps (blue) to reduce visual ambiguity. The final viewpoint (red) offers a significantly clearer perspective for the perception module, yielding a substantial improvement in perception score compared to the initial state.

the frozen perception module (e.g., detection confidence, segmentation quality, or 3D consistency). This enables operation in open-world settings where ground-truth annotations are unavailable, unlike prior active perception methods that either tightly couple exploration with specific architectures (Chaplot et al., 2021; Kotar & Mottaghi, 2022; Ding et al., 2023) or collect labeled data for retraining (Scarpellini et al., 2024; Jing & Kong, 2023). Moreover, unlike recent RL-based approaches that assume closed-set tasks or rely on task-specific uncertainty models (Ding et al., 2023; Fan et al., 2024), our framework leverages semantic reasoning to handle open-ended, natural-language-driven objectives (e.g., "Locate the bicycle next to the door.") while generalizing across diverse perception backbones.

To realize this vision, we transform a vision-language model (VLM) into a low-level pose controller through a two-stage training pipeline. First, we align the VLM with spatial reasoning via supervised fine-tuning on rule-based exploration trajectories that systematically probe indoor environments. Second, we refine the policy using unsupervised reinforcement learning (RL), where rewards are constructed from the outputs of the frozen perception module without access to any downstream perceptual annotations (e.g., ground-truth masks or bounding boxes). This decouples perception from control, creating a modular framework that seamlessly accommodates diverse off-the-shelf perception architectures without the need for task-specific retraining.

We validate our approach on three mainstream vision tasks on datasets ReplicaCAD (Szot et al., 2021) and HM3D (Ramakrishnan et al., 2021) in photo-realistic Habitat (Savva et al., 2019) including: visual grounding, segmentation, and 3D bounding box estimation. Sea[2] improves performance on ReplicaCAD and HM3D by 13.54%, 15.92%, and 27.68% and 22.16%, 12.49%, and 9.31%, respectively,

demonstrating that strategic viewpoint selection can recover domain-gap-induced degradation without a single annotated label. This establishes a new direction for label-efficient domain adaptation in embodied AI.

Our contributions can be summarized as follows:

- We present the first VLM-based active perception framework that achieves plug-and-play compatibility with diverse off-the-shelf models. By using only scalar outputs as rewards, Sea[2] enables seamless integration across various perception architectures without requiring retraining or downstream labels.

- We introduce an unsupervised RL training pipeline based on perception-derived rewards. By leveraging only task-level objectives and scalar outputs from frozen models, our method eliminates the need for dense perceptual annotations (e.g., pixel-level masks or 3D boxes), enabling effective policy learning in annotation-scarce environments.

- We demonstrate substantial gains across three visual tasks, including detection, segmentation, and 3D understanding, with improvements of 13.54%, 15.92%, and 27.68% in metrics on dataset ReplicaCAD, showing that viewpoint adaptation alone can effectively bridge domain gaps.

## 2. Related Work

### 2.1. Embodied Active Perception

Active perception studies how agents move to acquire views that improve recognition under occlusion and partial visibility. Early *Embodied Visual Recognition (EVR)* work

showed that learned motion policies can outperform passive baselines for classification, amodal localization, and segmentation in 3D environments (Yang et al., 2019). Subsequent lines either adapt the detector in-the-loop during interaction (Kotar & Mottaghi, 2022), or keep perception frozen and optimize the *policy* to seek informative views, e.g., decision-transformer control with offline+online training (Ding et al., 2023). Complementary efforts collect/evaluate informative multi-view data for self-training or exploration-aware adaptation (Fang et al., 2020; Chaplot et al., 2021; Jing & Kong, 2023; Scarpellini et al., 2024), incorporate uncertainty to guide viewpoint selection (Fan et al., 2024), extend from detection to active localization (Di Giammarino et al., 2024), and address embodied domain adaptation under distribution shift (Shi et al., 2025). Our work (i) keeps all perception modules *frozen* to avoid catastrophic forgetting, (ii) uses a single VLM policy that conditions on both images and natural-language task prompts to *control viewpoints* across multiple modules.

## 2.2. Multimodal Foundation Models

Multimodal foundation models provide open-vocabulary recognition and promptable perception. CLIP (Radford et al., 2021) aligns images and text for robust zero-shot recognition; BLIP-2 (Li et al., 2023) bridges frozen vision encoders and LLMs for efficient multimodal reasoning; Flamingo scales few-shot VLMs; instruction-tuned MLLMs such as LLaVA (Liu et al., 2023) improve visual instruction following. Recent families (Qwen3-VL (Yang et al., 2025), InternVL3.5 (Wang et al., 2025)) further enhance grounding and OCR across parameter scales. Recent work has utilized the multimodal perception and planning capabilities of VLM for embodied navigation (Zhang et al., 2024b;a; Jiang et al., 2025; Ao et al., 2025; Yin et al., 2024; Zhang et al., 2025) rather than purely for perception or generation. Inspired by this, we *use a VLM as an action policy* to translate language-grounded spatial reasoning into low-level control while keeping task modules frozen.

## 2.3. RL-based Post-training for VLMs

RLHF learns a reward model from human preferences, then optimize the policy with PPO under a KL constraint (InstructGPT) (Ouyang et al., 2022); strong alignment but costly and reward-sensitive. DPO removes reward modeling and directly fit a preference objective from pairs (Rafailov et al., 2023); simpler but still preference-dependent. GRPO-style objectives have recently improved long-horizon reasoning and training stability (Shao et al., 2024; Zheng et al., 2025; Yu et al., 2025). We adopt GRPO to train the VLM as an action policy from *unsupervised* rewards avoiding preference data and detector fine-tuning while enabling transferable, language-guided active perception.

## 3. Method

Our core premise is that perceptual performance degradation in novel domains stems not only from model capacity limitations, but also from suboptimal viewpoint selection (Yang et al., 2019; Fan et al., 2024). Rather than adapting perception modules, which risks catastrophic forgetting and demands costly annotations, we freeze all perception models and instead learn an unsupervised pose-control policy that navigates toward informative observations. The policy is optimized solely through scalar feedback from the frozen modules (confidence scores and geometric consistency), requiring no downstream labels during training. In this section, we introduce our framework design, which is an indoor VLM-guided embodied agent with three perception modules. We also elaborate on a two-stage training pipeline for our proposed method.

### 3.1. Overview and Problem Formulation

We formulate the task as an unsupervised active perception problem where an embodied agent controls its camera pose to maximize the quality of observations for a set of frozen perception modules $\mathcal{M}$. At episode onset, the agent receives a natural-language task instruction $I$ and an initial observation $o_1$. The agent's policy $\pi_\theta$ generates the task prompt $p$, selects the perception module $m \in \mathcal{M}$ and outputs discrete actions $a_t \in \mathcal{A}$ over a horizon $T$ to adjust its viewpoint, where $\mathcal{A}$ includes translational and rotational movements (e.g., move forward, turn right/left, look up/down, stop).

The observation $o_t$ at time $t$ is an RGB image captured from the current pose. The frozen selected perception module $m$ processes $(o_t, p)$ to produce a prediction $\hat{y}_t^m$ and a scalar confidence score $c_t^m \in [0, 1]$ indicating result reliability. Critically, no ground-truth labels are available for $\hat{y}_t^m$; the agent must infer viewpoint quality solely from the sequence of $(\hat{y}_t^m, c_t^m)$.

**Optimization Objective.** The agent's goal is to learn a policy $\pi_\theta$ that maximizes the cumulative quality of perception outcomes without updating any module parameters:

$$\max_\theta \mathbb{E}_{\pi_\theta} \left[ \sum_{t=1}^T r\left(\{\hat{y}_t^m, c_t^m\}_{m\in\mathcal{M}}, o_t\right) \right] \quad (1)$$

where the reward $r(\cdot)$ is an unsupervised scalar signal derived entirely from the frozen modules' outputs and geometric consistency checks. This decouples perception from control, enabling zero-shot transfer across module architectures and scenes.

### 3.2. Framework Design

Sea² employs a VLM as the action policy $\pi_\theta$. The VLM receives a task instruction $I$ (encoding the original task type

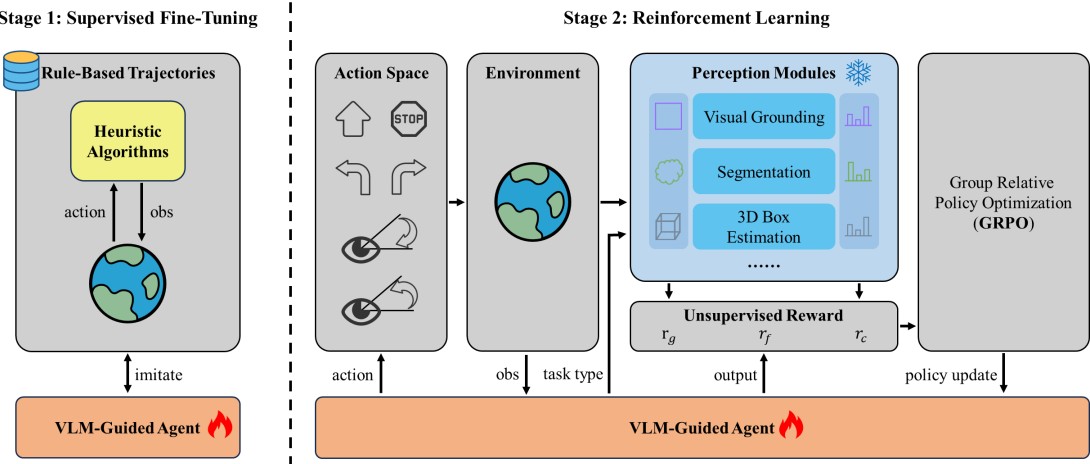

*Figure 3.* **Illustration of our Sea$^2$ framework.** In Stage 1, the VLM is fine-tuned on rule-based trajectories generated by heuristic algorithms to align it with spatial reasoning and control formats. In Stage 2, the VLM serves as a low-level pose controller for the agent, where it is further refined using unsupervised reinforcement learning with GRPO. The agent interacts with the environment, receiving observations and taking actions to optimize its policy based on rewards derived from the frozen selected perception module's confidence and results (e.g., grounding confidence, mask area). The selected perception module remain frozen throughout the training process, ensuring no catastrophic forgetting of prior knowledge. The final policy enables the agent to navigate to informative viewpoints that enhance the performance of the perception modules without requiring any downstream annotations.

$h_I$ and language description $p_I$ of the target object, further details regarding the formulation of $I$ are provided in the **Task Definition** (see 4.1).) and the current observation $o_t$, then generates a structured output comprising:

- Thoughts: A textual reasoning trace explaining the spatial reasoning (e.g., object location, occlusion assessment).

- Task type: A task routing result $h$ used to select perception module $m \in \mathcal{M}$.

- Task prompt: A language description $p$ of the task for the selected perception module $m$.

- Action: A discrete control command $a_t \in \mathcal{A}$.

The VLM is transformed from a passive reasoning model into an embodied pose controller through our two-stage training pipeline, *i.e.*, supervised fine-tuning (SFT) followed by reinforcement learning (RL).

**Perception Module Interface.** Each module $m \in \mathcal{M}$ implements a unified interface: given $(o_t, p)$, it returns a prediction $\hat{y}_t^m$ (*e.g.*, grounded 2D box, segmentation mask, or 3D box) and a confidence score $c_t^m$. The confidence is based on each perception module. During both SFT and RL, all modules remain frozen, treating them as black-box experts that provide feedback without adaptation.

### 3.3. Training Pipeline

As shown in Figure 3, the first stage involves supervised fine-tuning (SFT) of the VLM using rule-based trajectories. The second stage employs reinforcement learning (RL) with reward signals derived from perception modules' feedback.

#### 3.3.1. SUPERVISED FINE-TUNING (SFT)

To initialize the VLM with basic spatial reasoning and reduce RL exploration variance, we collect trajectories using a deterministic heuristic policy. The heuristic follows a three-phase logic:

- Object Search: Rotate until the target object is detected with non-zero confidence.

- Viewpoint Centering: Adjust the viewpoint to align the object's predicted region with the image center.

- Proximity Adjustment: Move forward until the object occupies a sufficient image area, then stop.

Each trajectory records the heuristic's "thoughts" (spatial reasoning), task type, task prompt and actions, forming a supervised dataset for SFT that aligns the VLM's output format with embodied control requirements.

#### 3.3.2. REINFORCEMENT LEARNING (RL)

After acquiring fundamental embodied perception skills through SFT, we further improve the model via reinforcement learning (RL) using perception-based reward signals.

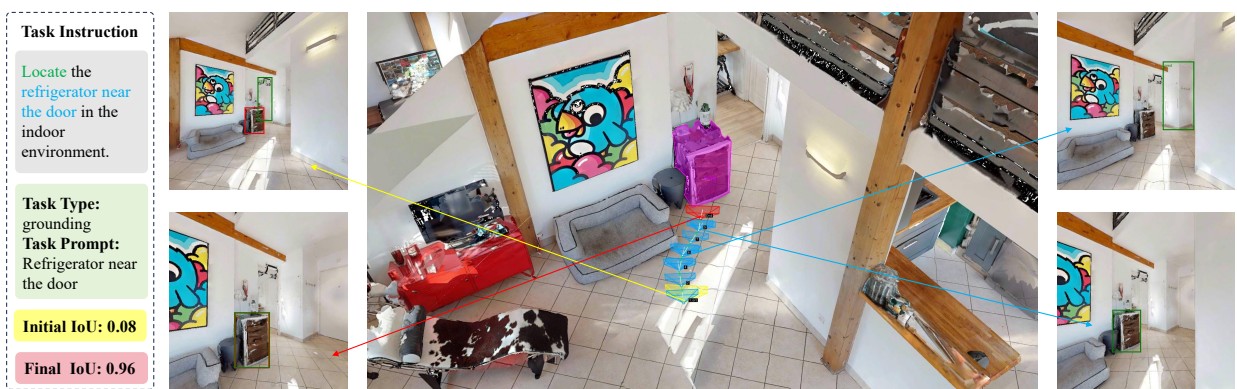

*Figure 4.* **Illustration of the active perception process for visual grounding.** From a poor initial view (yellow) where the prediction (green box) for the target (red box) is inaccurate, the agent takes navigational steps (blue) to reduce ambiguity, reaching a final viewpoint (red) that greatly improves the perception result.

In this stage, we adopt the Group Relative Policy Optimization (GRPO) algorithm and design a comprehensive reward function to guide policy optimization. The total reward is composed of a *format reward* $r_f$, a *confidence reward* $r_c$, and a *geometric reward* $r_g$. Among them, $r_f$ is derived from the action policy output, while $r_c$ and $r_g$ are computed from the perception module outputs. The overall reward function is defined as:

$$r = \begin{cases} r_f + \lambda_1 r_c + \lambda_2 r_g, & \text{if } h = h_I \\ -1, & \text{if } h \neq h_I \end{cases} \quad (2)$$

where $h$ and $h_I$ denote the predicted and ground-truth task types, respectively. The agent receives a constant penalty of $-1$ for incorrect task identification; otherwise, the reward is defined as a weighted combination of the fundamental perception score $r_f$, the confidence score $r_c$, and the geometric consistency $r_g$. $\lambda_1$ and $\lambda_2$ weight the respective rewards and satisfy $\lambda_1 + \lambda_2 = 1$. Specifically, while the agent utilizes the ground-truth task type $h_I$ to determine the reward structure, the training process remains entirely independent of perceptual ground-truth annotations. This ensures that the policy learns to optimize observation quality using only the intrinsic feedback of the frozen perception models, making it applicable to open-world scenarios where precise spatial or semantic labels are unavailable.

**Format Reward.** The format reward $r_f$ evaluates whether the model's output structure adheres to the expected schema. Specifically, the reasoning process must be contained in the `"thoughts":{}` field, the action decision must appear in the `"action":{}` field, and only one action can be generated per step. This reward encourages the model to output well-structured and consistent reasoning–action pairs:

$$r_f = \begin{cases} 0.05, & \text{if the format is correct,} \\ -0.05, & \text{if the format is incorrect.} \end{cases} \quad (3)$$

**Confidence Reward.** The confidence reward $r_c$ measures the change in the perception module's confidence score between consecutive steps, encouraging the agent to increase confidence in its perception results without relying on external supervision. For visual grounding, the confidence is obtained from GroundingDINO; for segmentation, it is a weighted combination of GroundingDINO and SAM; and for 3D box estimation, it integrates confidence from GroundingDINO, SAM, and the 3D box estimator.

$$r_c = c_t^m - c_{t-1}^m \quad (4)$$

**Geometric Reward.** The geometric reward $r_g$ enforces spatial consistency between the predicted region and the observation. It consists of two components: *area* and *center*.

The area reward measures the proportion of the predicted region $\hat{y}_t^m$ within the current observation $o_t$, encouraging the model to approach the target object. For visual grounding, this ratio corresponds to the bounding-box area relative to the image; for segmentation, to the mask area; and for 3D box estimation, to the projected 3D box area.

$$r_a = \frac{A(\hat{y}_t^m)}{A(o_t)} \quad (5)$$

where $A(\cdot)$ is the area function.

The center reward encourages the model to align the predicted target region with the image center. Instead of using raw Euclidean distance, we compute a normalized alignment score between 0 and 1, where higher values indicate better alignment. This design stabilizes localization behavior and prevents excessive deviation from the target region during training.

$$r_u = 1 - d\big(u(\hat{y}_t^m), u(o_t)\big) \quad (6)$$

where $u(\cdot)$ extracts the predicted and observed center coordinates and $d(\cdot, \cdot)$ computes normalized distance.

*Table 1.* Three perception module baselines on ReplicaCAD with different controllers. Visual grounding is evaluated by mAP, segmentation by IoU and Dice, and 3D box estimation by IoU and Center Score (higher is better). The best results are highlighted in **bold**. Degraded results are highlighted in red, while improved results are highlighted in green. **Notably, the Shortest Path baseline is a ground-truth-informed (oracle) method: it possesses perfect prior knowledge of target locations, retrieving precise 3D coordinates from scene annotations and computing the shortest navigable path using the Habitat pathfinder.**

| | Visual Grounding | | | Segmentation | | 3D Box Estimation | |
|---|---|---|---|---|---|---|---|
| **Perception Module + Policy** | mAP@0.5 ↑ | mAP@0.75 ↑ | mAP$_{avg}$ ↑ | IoU ↑ | Dice ↑ | IoU ↑ | Center Score ↑ |
| Pretrained Perception Module (PPM) | 0.7958 | 0.6225 | 0.7092 | 0.5621 | 0.6398 | 0.2648 | 0.5499 |
| PPM + Forward | 0.4667 | 0.3897 | 0.4282 | 0.3495 | 0.3911 | 0.1503 | 0.3206 |
| | (-41.37%) | (-37.40%) | (-39.62%) | (-37.82%) | (-38.87%) | (-43.24%) | (-41.70%) |
| PPM + Random | 0.6662 | 0.5183 | 0.5923 | 0.4626 | 0.5282 | 0.2237 | 0.4809 |
| | (-16.29%) | (-16.74%) | (-16.48%) | (-17.70%) | (-17.44%) | (-15.48%) | (-12.55%) |
| PPM + Heuristic | 0.6969 | 0.5698 | 0.6334 | 0.5200 | 0.5849 | 0.2543 | 0.5310 |
| | (-12.43%) | (-8.47%) | (-10.69%) | (-7.49%) | (-8.58%) | (-3.93%) | (-3.44%) |
| PPM + Shortest Path | 0.8668 | 0.7198 | 0.7933 | 0.6075 | 0.6817 | 0.3259 | 0.6373 |
| | (+8.92%) | (+15.64%) | (+11.86%) | (+8.07%) | (+6.55%) | (+23.09%) | (+15.89%) |
| PPM + InternVL3.5-2B | 0.7020 | 0.5747 | 0.6384 | 0.5556 | 0.6322 | 0.2528 | 0.5250 |
| | (-11.79%) | (-7.68%) | (-9.99%) | (-1.16%) | (-1.19%) | (-4.53%) | (-4.52%) |
| PPM + Qwen3VL-2B | 0.6133 | 0.5097 | 0.5615 | 0.4214 | 0.4744 | 0.1781 | 0.4038 |
| | (-22.93%) | (-18.13%) | (-20.83%) | (-25.03%) | (-25.85%) | (-32.74%) | (-26.57%) |
| **PPM + Ours (InternVL3.5-2B)** | 0.8627 | 0.7055 | 0.7841 | 0.6251 | 0.7002 | 0.3195 | 0.6600 |
| | (+8.40%) | (+13.33%) | (+10.56%) | (+11.21%) | (+9.44%) | (+20.67%) | (+20.03%) |
| **PPM + Ours (Qwen3VL-2B)** | **0.8725** | **0.7378** | **0.8052** | **0.6516** | **0.7267** | **0.3380** | **0.6893** |
| | (+9.63%) | (+18.53%) | (+13.54%) | (+15.92%) | (+13.59%) | (+27.68%) | (+25.35%) |

The geometric reward $r_g$ calculation method is as follows:

$$r_g = g_t^m - g_{t-1}^m \tag{7}$$
$$g_t^m = \mu_1 r_a + \mu_2 r_u \tag{8}$$

where $g_t^m$ is the geometric score and $\mu_1$ and $\mu_2$ weight the respective rewards and satisfy $\mu_1 + \mu_2 = 1$.

# 4. Experiments

## 4.1. Experimental Setup and Data

**Task Definition.** At the beginning of each episode, the agent is spawned at a random navigable pose with a forward-facing RGB camera (height 1 m, 512×512 resolution). We select as target an object that is (partially) in view and exceeds a minimum visible-area threshold to avoid degenerate language references. The task instruction $I$ is synthesized via a two-stage hierarchical process. First, we employ Qwen2.5-VL-7B (Bai et al., 2025) to generate an open-vocabulary, attribute-aware language description $p_I$ of the target object. Subsequently, the final task instruction $I$ is formulated by conditioning on the ground-truth task type $h_I$ (e.g., "segment the $[p_I]$"), thereby encoding both the target's semantic attributes and the specific perceptual objective. The policy is allowed up to $T = 10$ steps and may issue stop early once it believes the current viewpoint is informative. Formally, given observation $o_t$ and prompt $p$, the VLM-based policy $\pi$ chooses an action $a_t = \pi(o_t, p)$ and we record the perception output sequence $(\hat{y}^m, c^m)$ for each frozen module $m \in \mathcal{M}$.

**Environments and Splits.** All experiments are conducted in Habitat (Savva et al., 2019) on ReplicaCAD (Szot et al., 2021) suite (84 indoor scenes) and HM3D (Ramakrishnan et al., 2021) suite. For each suite, we randomly select 10 scenes for training and sample 500 tasks from the 10 training scenes to construct the test sets for evaluation.

**Action Space and Observations.** We use a low-level discrete action set with step sizes of 0.25 m (translation) and 10° (yaw/pitch). Vertical look controls are included to handle tall/near objects and reduce self-occlusion. Each step provides the current RGB frame; depth is used only by the 3D module (below), not as an input channel to the policy.

**Perception Modules.** We consider three representative perception tasks and their modules $\mathcal{M}=\{$Visual Grounding, Segmentation, 3D Box Estimation$\}$; each returns a prediction and a scalar confidence $conf_m$ used for feedback and rewards. *(i) Visual Grounding* (VG) uses GroundingDINO (Liu et al., 2024) (GDINO) to produce a 2D bounding box and confidence $(\hat{y}^{vg}, c^{vg})$ given the language description and the current RGB frame. *(ii) Segmentation* first applies GroundingDINO to obtain a box and then SAM (Kirillov et al., 2023) to produce a mask $(\hat{y}^{seg})$. Its confidence is a weighted combination of detector and mask confidences $c^{seg}=\mu_3 c^{seg1}+\mu_4 c^{seg2}$ (fixed weights across experiments). *(iii) 3D Box Estimation* computes a point cloud by intersecting the mask with the depth image and fits an oriented 3D bounding box via minimum-area footprint search on the ground plane. Its confidence aggregates detector, mask,

*Table 2.* Ablation study on training strategies. Visual grounding is evaluated by mAP, segmentation by IoU and Dice, and 3D box estimation by IoU and Center Score (higher is better). The best results are highlighted in **bold**. Degraded results are highlighted in red, while improved results are highlighted in green.

| Method | Visual Grounding | | | Segmentation | | 3D Box Estimation | |
|---|---|---|---|---|---|---|---|
| | mAP@0.5 ↑ | mAP@0.75 ↑ | mAP$_{avg}$ ↑ | IoU ↑ | Dice ↑ | IoU ↑ | Center Score ↑ |
| Pretrained Perception Module | 0.7958 | 0.6225 | 0.7092 | 0.5621 | 0.6398 | 0.2648 | 0.5499 |
| RL-Only | 0.4647 | 0.3877 | 0.4262 | 0.3513 | 0.3921 | 0.1511 | 0.3239 |
| | (-41.61%) | (-37.72%) | (-39.90%) | (-37.50%) | (-38.71%) | (-42.92%) | (-41.09%) |
| SFT-Only | 0.8027 | 0.6738 | 0.7383 | 0.5997 | 0.6719 | 0.3099 | 0.6012 |
| | (+0.86%) | (+8.25%) | (+4.10%) | (+6.70%) | (+5.02%) | (+17.05%) | (+9.32%) |
| **SFT+RL** | **0.8627** | **0.7055** | **0.7841** | **0.6251** | **0.7002** | **0.3195** | **0.6600** |
| | (+8.40%) | (+13.33%) | (+10.56%) | (+11.21%) | (+9.44%) | (+20.67%) | (+20.03%) |

*Table 3.* Ablation study on reward functions. Visual grounding is evaluated by mAP, segmentation by IoU and Dice, and 3D box estimation by IoU and Center Score (higher is better). Degraded results are highlighted in red, while improved results are highlighted in green.

| Reward function | Visual Grounding | | | Segmentation | | 3D Box Estimation | |
|---|---|---|---|---|---|---|---|
| | mAP@0.5 ↑ | mAP@0.75 ↑ | mAP$_{avg}$ ↑ | IoU ↑ | Dice ↑ | IoU ↑ | Center Score ↑ |
| Format | — | — | — | — | — | — | — |
| Format + Geometric | 0.8292 | 0.6872 | 0.7582 | 0.5942 | 0.6667 | 0.2905 | 0.6125 |
| | (+4.20%) | (+10.39%) | (+6.91%) | (+5.71%) | (+4.20%) | (+9.71%) | (+11.38%) |
| Format + Confidence | 0.7115 | 0.5562 | 0.6339 | 0.5253 | 0.5973 | 0.2902 | 0.5954 |
| | (-10.60%) | (-10.65%) | (-10.62%) | (-6.54%) | (-6.64%) | (+9.59%) | (+8.27%) |
| Format + Confidence + Geometric | 0.8627 | 0.7055 | 0.7841 | 0.6251 | 0.7002 | 0.3195 | 0.6600 |
| | (+8.40%) | (+13.33%) | (+10.56%) | (+11.21%) | (+9.44%) | (+20.67%) | (+20.03%) |

and geometric signals (e.g., point count, filtering retention, depth consistency) into $c^{3D} \in [0, 1]$.

**SFT Data Collection.** To align the pretrained VLM with embodied control and reduce RL exploration burden, we collect supervised trajectories with a deterministic heuristic that is agnostic to perception architectures: *Search*—rotate until the target becomes visible; *Centering*—partition the image into a 3×3 grid; if the target center is off-axis, issue {turn_left, turn_right, look_up, look_down} to re-center; *Approach/Stop*—once centered, if the visible region is below a threshold, move_forward; otherwise stop. At each step we log the textual "thoughts" and the "action" token. These trajectories seed the *SFT* stage.

**GRPO Training Details.** For reinforcement learning, we optimize the policy with GRPO using the following hyperparameters: initial learning rate $1 \times 10^{-6}$, 4 rollouts per update, batch size 8, KL coefficient 0.04, and sampling temperature 0.4. Rewards are computed from frozen-module feedback as defined in Equation (2) (confidence change in Equation (4) and geometric shaping in Equation (6)).

**Baselines and Metrics.** We compare against: *(a) Pretrained Perception Module (PPM)*—evaluate each frozen module on the first frame at $t$=1 (no action); *(b) Forward*—always execute move_forward; *(c) Random*—sample a random action from $\mathcal{A}$ at each step; *(d) Heuristic*—the

search–centering–approach heuristic used for SFT data collection; *(e) Shortest Path*—a baseline that has access to ground-truth object positions, serving as a reference for evaluating learned policies under the same constraints.

For baselines (a)–(e), the ground-truth task type $h_I$ and target description $p_I$ are directly provided as inputs. In contrast, for *(f) InternVL3.5-2B (Wang et al., 2025)* & *Qwen3VL-2B (Yang et al., 2025)* (prompt-based VLM controllers) and *(g) Ours (InternVL3.5)* & *Ours (Qwen3VL)* (trained via our two-stage pipeline), the models receive only the natural language instruction $I$. These models must autonomously parse the task type $h$ and object description $p$ from $I$. To ensure rigorous evaluation, if a model misidentifies the task type ($h \neq h_I$), the corresponding metric for that episode is set to zero; furthermore, any distortion in the extracted $p$ naturally impacts the downstream perception module's performance.

We use AP for visual grounding, IoU/Dice for segmentation, and IoU/center score for 3D box estimation. Metrics are computed from the final module outputs at stop (or $t$=$T$). Baselines and metrics correspond to Table 1.

### 4.2. Performance in Perception Tasks

Following the setup and baselines defined in Section 4.1, Table 1 summarizes results on ReplicaCAD across visual grounding, segmentation, and 3D box estimation with frozen modules. Relative to the PPM baseline, our policy

*Table 4.* Comparison with `TTAOD-Foundation` (Gao et al., 2026) on visual grounding. Results are reported separately from the main baseline table because this baseline uses the default `GroundingDINO` version supported by its public implementation. Gains are computed against `PPM` under the same detector setting.

| Dataset | Method | mAP@0.5 ↑ | mAP@0.75 ↑ | mAP$_{avg}$ ↑ |
|---|---|---|---|---|
| ReplicaCAD | PPM | 0.8181 | 0.5708 | 0.6945 |
| | TTAOD-Foundation | 0.8389 (+2.54%) | 0.6008 (+5.26%) | 0.7199 (+3.66%) |
| | **PPM + Ours** | **0.8675** (+6.04%) | **0.6975** (+22.20%) | **0.7825** (+12.67%) |
| HM3D | PPM | 0.6214 | 0.4320 | 0.5267 |
| | TTAOD-Foundation | 0.6776 (+9.04%) | 0.4721 (+9.28%) | 0.5749 (+9.14%) |
| | **PPM + Ours** | **0.6799** (+9.41%) | **0.5605** (+29.75%) | **0.6202** (+17.75%) |

*Table 5.* Analysis of confidently wrong predictions across different confidence regimes.

| Task | Setting | $\tau$ | Correct Rate ↑ | Improve Rate ↑ | Avg $\Delta$IoU ↑ |
|---|---|---|---|---|---|
| Visual Grounding | High-Confidence Wrong | 0.6 | 18/41 (43.90%) | 26/41 (63.41%) | +0.2403 |
| | | 0.7 | 13/26 (50.00%) | 17/26 (65.38%) | +0.2489 |
| | | 0.8 | 8/15 (53.33%) | 9/15 (60.00%) | +0.1833 |
| | Low-Confidence Wrong | 0.6 | 61/86 (70.93%) | 66/86 (76.74%) | +0.4695 |
| | | 0.7 | 66/101 (65.35%) | 75/101 (74.26%) | +0.4333 |
| | | 0.8 | 71/112 (63.39%) | 83/112 (74.11%) | +0.4239 |
| Segmentation | High-Confidence Wrong | 0.7 | 67/203 (33.00%) | 136/203 (67.00%) | +0.1974 |
| | Low-Confidence Wrong | 0.7 | 18/43 (41.86%) | 31/43 (72.09%) | +0.2912 |
| 3D Box Estimation | High-Confidence Wrong | 0.7 | 52/313 (16.61%) | 209/313 (66.77%) | +0.0899 |
| | Low-Confidence Wrong | 0.7 | 13/72 (18.06%) | 46/72 (63.89%) | +0.1933 |

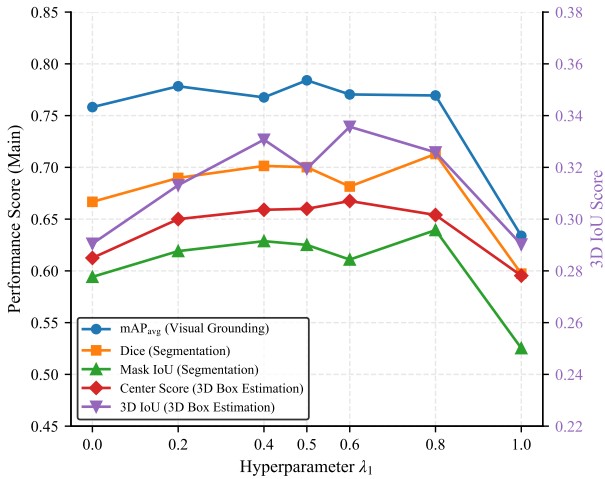

*Figure 5.* Impact of hyperparameter $\lambda_1$ on different tasks. The left y-axis corresponds to Grounding and Segmentation metrics, while the right y-axis highlights the 3D IoU performance.

*Table 6.* Impact of hyperparameter $\mu_1$ on different tasks.

| $\mu_1$ | Grounding (mAP$_{avg}$) | Seg. (IoU) | 3D Box (IoU) |
|---|---|---|---|
| 0.00 | 0.6975 | 0.5515 | 0.2944 |
| 0.25 | 0.7684 | 0.6033 | 0.2863 |
| 0.50 | 0.7503 | 0.5841 | 0.3140 |
| **0.75** | **0.8052** | **0.6516** | **0.3380** |
| 1.00 | 0.7336 | 0.5725 | 0.2942 |

quire matters. The `Heuristic` baseline improves segmentation and 3D box metrics only marginally (e.g., +3.37% seg IoU) and lags significantly behind our policy. This is primarily because `Heuristic` is purely reactive, relying entirely on the immediate feedback of the frozen perception module; consequently, any initial misdetection becomes unrecoverable, leading the agent into erroneous trajectories. Meanwhile, although `Shortest path` possesses privileged knowledge of the target's 3D coordinates to ensure geometric reachability, its performance gains remain modest (e.g., +6.27% 3D IoU). This indicates that simply arriving at the target's location is insufficient for high-quality perception. Unlike `Shortest path`, which lacks the capacity for viewpoint-level planning, Sea$^2$ strategically optimizes the camera pose to mitigate occlusion and maximize visual informativeness. Finally, directly prompting a compact VLM controller without task-aligned training underperforms the static initial value on all metrics, highlighting that naïvely using a VLM as a policy is insufficient; embodiment alignment and reward-driven refinement are

boosts VG AP$_{avg}$ +13.54%, segmentation IoU +15.92% with Dice rising +13.59%, and 3D box IoU +27.68%, 3D center score +25.35%.

Simple motion baselines are ineffective or even harmful. `Forward` policy substantially degrades all metrics (e.g., −41.37% seg IoU), likely due to over-approaching that increases occlusion or truncation. `Random` motion is also inferior across tasks, underscoring that *which* view to ac-

necessary.

We further evaluate Sea$^2$ on HM3D, a dataset featuring more complex and high-fidelity 3D-scanned reconstructions of real-world environments. As summarized in Table 7, the performance gains on HM3D mirror those observed on ReplicaCAD, reinforcing the effectiveness and robustness of our active perception strategy across diverse scene distributions.

We further compare against `TTAOD-Foundation` (Gao et al., 2026), a recent test-time adaptation baseline for foundation-model-based detection. For fairness, we train it with a dataset of the same scale as that used for our visual-grounding policy training and evaluate all methods with the default `GroundingDINO` version supported by its public implementation; we therefore report the results separately in Table 4. Sea$^2$ consistently outperforms `TTAOD-Foundation` on both ReplicaCAD and HM3D, with especially large gains at mAP@0.75 and mAP$_{avg}$, indicating that active viewpoint adaptation improves localization quality more effectively than parameter adaptation alone.

A potential concern is that confidence-based rewards may be unreliable under domain shift: frozen perception modules can be miscalibrated on OOD observations and may produce high-confidence but incorrect predictions. In that case, an RL agent might learn to move toward viewpoints that reinforce these errors rather than resolve them. To mitigate this risk, our reward is not based on confidence alone, but also includes geometric consistency terms. We further test this issue directly in Table 5 by separating initially incorrect samples ($initial\_iou < 0.5$) into a high-confidence wrong (HCW) subset with $initial\_conf \geq \tau$ and a low-confidence wrong (LCW) control subset with $initial\_conf < \tau$. Here, Correct Rate denotes the fraction of cases with $final\_iou \geq 0.5$, Improve Rate denotes the fraction with $final\_iou > initial\_iou$, and Avg $\Delta$IoU is the mean change $final\_iou - initial\_iou$. Even on HCW samples, the agent still improves predictions in 65.38% of visual-grounding cases at $\tau=0.7$, 67.00% of segmentation cases, and 66.77% of 3D box cases. These results suggest that the policy does not simply reinforce miscalibrated confidence, even when the initial prediction is confidently wrong.

### 4.3. Ablation Study

**Training Strategy.** Table 2 compares *RL-Only*, *SFT-Only*, and our *SFT+RL*. Training the policy from scratch with RL is unstable and underperforms the initial value on recognition metrics. SFT on rule-based trajectories yields consistent gains across tasks, validating the importance of a spatially grounded cold start. The full *SFT+RL* pipeline achieves the best results on all metrics, showing that SFT provides stable priors while RL unlocks task-specific view selection beyond heuristic behaviors.

**Reward Design.** Table 3 further disentangles the contribution of each reward term in Equation (2). Using *confidence* alone (*Format+Confidence*) is unstable: it degrades visual grounding and segmentation, while offering improvements only on 3D box estimation. Although confidence is a strong directional signal toward better viewpoints, it is also noisy and volatile; the effect is most pronounced in VG, where the signal comes solely from `GroundingDINO`, making the policy more susceptible to detector-specific jitter. By contrast, the *geometric* term (*Format+Geometric*) is stable and module-agnostic. Optimizing area/center alignment (Equation (6)) yields consistent but limited improvements across tasks, reflecting that geometric shaping provides reliable spatial guidance yet lacks model/-task specificity. Crucially, combining the two signals (*Format+Confidence+Geometric*) produces large and uniform gains. Geometry supplies smooth, low-variance spatial shaping, while confidence calibrates view selection to the current frozen module and task, turning a noisy-but-informative signal into a robust control objective. Figure 5 analyzes the impact of $\lambda_1$, indicating a necessary trade-off between the reward components. We additionally study the geometric reward weight in Table 6; setting $(\mu_1, \mu_2)=(0.75, 0.25)$ yields the best overall trade-off across visual grounding, segmentation, and 3D box estimation. Consequently, we fix $\lambda_1 = 0.5$ and $(\mu_1, \mu_2)=(0.75, 0.25)$ for all main evaluations.

## 5. Conclusions

We presented our Sea$^2$ that adapts pre-trained perception modules to new embodied environments without updating parameters or requiring downstream labels. By learning a pose-control policy through supervised heuristic trajectories and unsupervised GRPO-based RL, the agent actively selects informative viewpoints using only scalar feedback from frozen perception modules. Experiments on visual grounding, segmentation, and 3D box estimation demonstrate gains, 13.54%, 15.92%, and 27.68% respectively on ReplicaCAD, highlighting that intelligently controlling viewpoints offers an efficient and annotation-free alternative to traditional model fine-tuning for domain adaptation.

**Limitations.** Keeping the perception modules frozen avoids catastrophic forgetting, but it also caps performance at the capability of the pre-trained models. Even with an optimal viewpoint, the system cannot recover errors caused by missing category knowledge or weak out-of-domain generalization in the underlying perception backbone. An important direction for future work is to combine active viewpoint selection with parameter-efficient adaptation, so that the system can retain the robustness benefits of frozen models while better addressing domain-specific failure cases.

## Acknowledgements

This work was supported by the National Natural Science Foundation of China (No. 62576308), Zhejiang Provincial Natural Science Foundation of China (No. LZ24F030005), and Fundamental Research Funds for the Central Universities (No. 226-2025-00167).

## Impact Statement

This work advances fundamental research in active perception and embodied AI by enabling frozen vision models to adapt to new environments through intelligent viewpoint selection without requiring labeled data or model updates. While our method could potentially improve the efficiency and generalization of vision systems in real-world applications (e.g., robotics, assistive technologies), it does not introduce novel ethical risks beyond those already present in general-purpose computer vision and language models. We do not anticipate immediate negative societal impacts, but as with any AI system deployed in physical environments, careful consideration of safety, robustness, and alignment with human intent would be necessary in downstream applications.

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

# A. 3D Box Estimation

The 3D box estimation module begins by using `GroundingDINO` to generate a bounding box and a confidence score for the target object based on the task and the current observation $o$. Next, `SAM` processes the observation $o$ and the box to produce a segmentation mask and its confidence score. The mask is then combined with the depth image to extract the point cloud of the target object. Finally, the point cloud is projected onto the horizontal plane, and a minimum bounding rectangle is computed by traversing all possible orientations of the projected points. The dimensions and orientation of this rectangle are used to construct the final 3D bounding box of the target object.

**3D Box Estimator.**  The 3D box estimator aims to robustly infer the spatial extent and orientation of the object from its segmented point cloud $P = \{p_i = (x_i, y_i, z_i)\}_{i=1}^{N}$. First, statistical filtering is applied to remove outliers:

$$P' = \{p_i \in P \mid \|p_i - \bar{p}\| < \tau\}, \tag{9}$$

where $\bar{p}$ is the mean of all points in $P$ and $\tau$ is a distance threshold. The filtered points $P'$ are then projected onto the horizontal plane:

$$P_{xy} = \{(x_i, y_i) \mid (x_i, y_i, z_i) \in P'\}. \tag{10}$$

We search over possible orientations $\theta$ to find the minimum-area bounding rectangle:

$$\theta^* = \arg\min_{\theta} \text{Area}\big(R(P_{xy}, \theta)\big), \tag{11}$$

where $R(P_{xy}, \theta)$ denotes the rectangle enclosing $P_{xy}$ after rotation by $\theta$. Finally, the 3D bounding box $B_{3D}$ is defined as:

$$B_{3D} = \big[\theta^*, w, l, h, c_x, c_y, c_z\big], \tag{12}$$

where $(w, l, h)$ are the width, length, and height of the box, and $(c_x, c_y, c_z)$ is its centroid.

**Confidence Computation.**  To evaluate the reliability of the estimated 3D box, we compute a geometric confidence score $c^{3D} \in [0, 1]$ based on three complementary criteria:

1. **Point Count Confidence:** Ensures a sufficient number of valid points for stable estimation:

$$C_{\text{points}} = \sigma\left(\frac{N' - N_0}{k}\right), \tag{13}$$

   where $N'$ is the number of filtered points, $N_0$ is a minimum threshold (e.g., 100), $k$ is a scaling factor, and $\sigma(\cdot)$ is the sigmoid function.

2. **Filtering Quality Confidence:** Measures the ratio of retained points after denoising:

$$C_{\text{quality}} = \frac{N'}{\max(1, N_{\text{orig}})}. \tag{14}$$

3. **Depth Consistency Confidence:** Evaluates the depth variance inside the mask:

$$C_{\text{depth}} = \frac{1}{1 + \frac{\sigma_d}{\tau_d}}, \tag{15}$$

   where $\sigma_d$ is the standard deviation of valid depth values and $\tau_d$ is a normalization constant.

The final geometric confidence is computed as a weighted combination of the three terms:

$$c^{3D} = w_1 C_{\text{points}} + w_2 C_{\text{quality}} + w_3 C_{\text{depth}}, \quad \text{with} \sum_i w_i = 1. \tag{16}$$

A higher $c^{3D}$ indicates greater geometric integrity and higher reliability of the estimated 3D bounding box. This confidence score is later incorporated into the confidence reward term $r_c$ during the RL training stage.

```
HABITAT_UNIFIED_COT_TEMPLATE = """<image>You are an embodied agent in an indoor
    environment. Your task is to navigate and improve the score of a visual perception
    task.

Task: {task_description}
Current Score: {conf_score:.3f} (range 0~1)

Based on the task description, you need to:
1. Determine what type of visual perception task this is:
    - grounding: Find a specific object described in natural language
    - segment: Segment a specific object described in natural language
    - 3d-box: Predict the 3D bounding box of an object
2. Extract the target object description (task_prompt) from the task description
3. Observe the current image and decide on an action

Available actions: ['move_forward', 'turn_left', 'turn_right', 'look_up', 'look_down',
    'stop']. You can only select 1 action at a time.

Your response should be a valid JSON in the following format:
{{
"thoughts": "{{First, analyze the task description to determine the task type and
    identify the target object. Then, describe where the target object is in the image
    (position, visibility, occlusion). Finally, decide which action to take.}}",
"task_type": "{{one of: grounding, segment, 3d-box}}",
"task_prompt": "{{a short phrase describing the target object, extracted from the task
    description}}",
"action": "{{action}}"
}}"""
```

*Listing 1.* The Unified Prompt Template used for all embodied perception tasks. This template guides the agent to perform task categorization and object extraction before action selection.

## B. Prompt Templates

In our experiments, we employ a unified prompt template for all visual perception tasks, including visual grounding, segmentation, and 3D bounding box estimation. This unified design requires the agent to not only decide on an action but also to explicitly identify the task type and extract the target object description from the natural language instruction.

As shown in Listing 1, the template follows a structured format that encourages Chain-of-Thought (CoT) reasoning. The agent receives the current RGB observation, a task description, and the current perception score. It then generates a JSON object containing its internal reasoning (`"thoughts"`), the identified `"task_type"`, the extracted `"task_prompt"`, and the selected `"action"`.

## C. Examples

We provide examples of three types of tasks. These examples illustrate how viewpoint adaptation reduces occlusion and scale ambiguity, leading to cleaner masks, tighter grounding, and more accurate 3D geometry.

## D. Additional Experiments

**Catastrophic Forgetting Under Downstream Fine-Tuning.** To directly test catastrophic forgetting under downstream fine-tuning, we fine-tuned `GroundingDINO` on 1,800 images collected from HM3D target scenes. As shown in Table 8, target-domain performance improves after fine-tuning, but source-domain COCO mAP drops sharply, supporting our design choice to keep the perception modules frozen.

**Catastrophic Forgetting Under TTA.** We further evaluate whether `TTAOD-Foundation` (Gao et al., 2026) also degrades source-domain performance. As shown in Table 9, although this baseline improves target-domain results in the

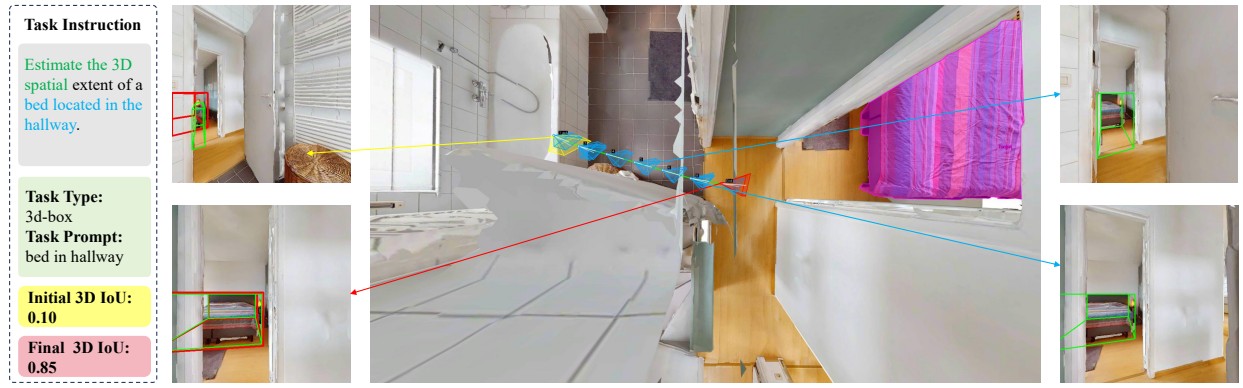

*Figure 6.* **Illustration of the active perception process for 3D box estimation.** From a poor initial view (yellow) where the prediction (green box) for the target (red box) is inaccurate, the agent takes navigational steps (blue) to reduce ambiguity, reaching a final viewpoint (red) that greatly improves the perception result.

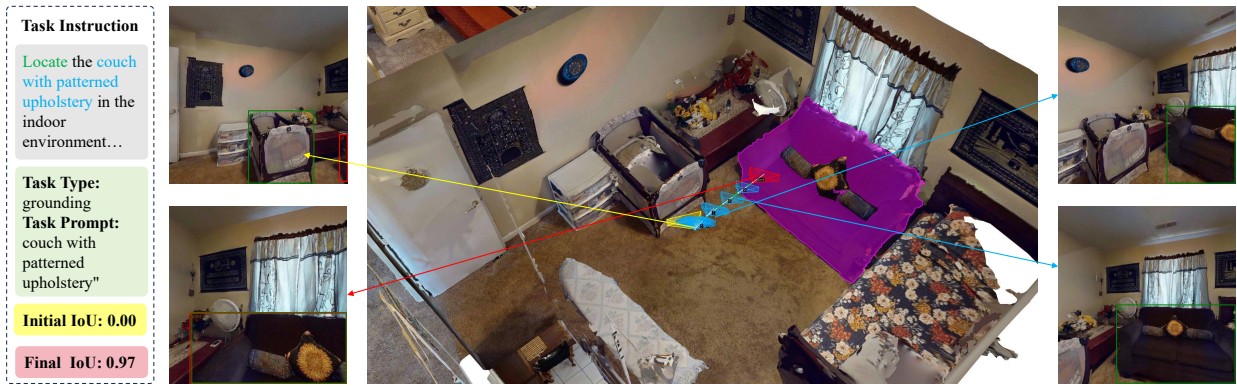

*Figure 7.* **Illustration of the active perception process for visual grounding.** From a poor initial view (yellow) where the prediction (green box) for the target (red box) is inaccurate, the agent takes navigational steps (blue) to reduce ambiguity, reaching a final viewpoint (red) that greatly improves the perception result.

main text, it reduces COCO mAP from 0.5031 to 0.4228, whereas our method preserves source performance because it adapts viewpoints without updating detector parameters.

**Robustness to Imperfect Depth Estimation.** To evaluate robustness to imperfect depth sensing, we inject Gaussian noise into the depth maps used by the 3D box estimator. As shown in Table 10, performance degrades gracefully as noise increases and remains clearly above the static baseline even at the largest noise level.

**Sensitivity to Initial Spatial Prediction Precision.** To assess how sensitive the geometric consistency check is to the initial spatial prediction, we perturb the initial 2D boxes, segmentation masks, and 3D boxes before computing the geometric rewards. The perturbation combines Gaussian center translation and log-normal size scaling, with three levels: Light $(0.02, 0.06)$, Medium $(0.04, 0.10)$, and Heavy $(0.08, 0.18)$. Results in Table 11 show that the policy remains above the static initial-viewpoint baseline across all noise levels.

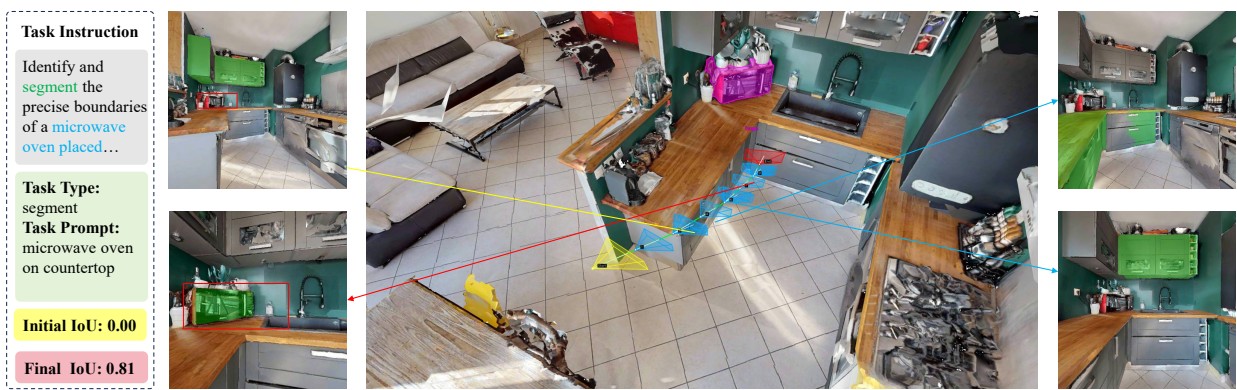

*Figure 8.* **Illustration of the active perception process for segmentation.** From a poor initial view (yellow) where the prediction (green box) for the target (red box) is inaccurate, the agent takes navigational steps (blue) to reduce ambiguity, reaching a final viewpoint (red) that greatly improves the perception result.

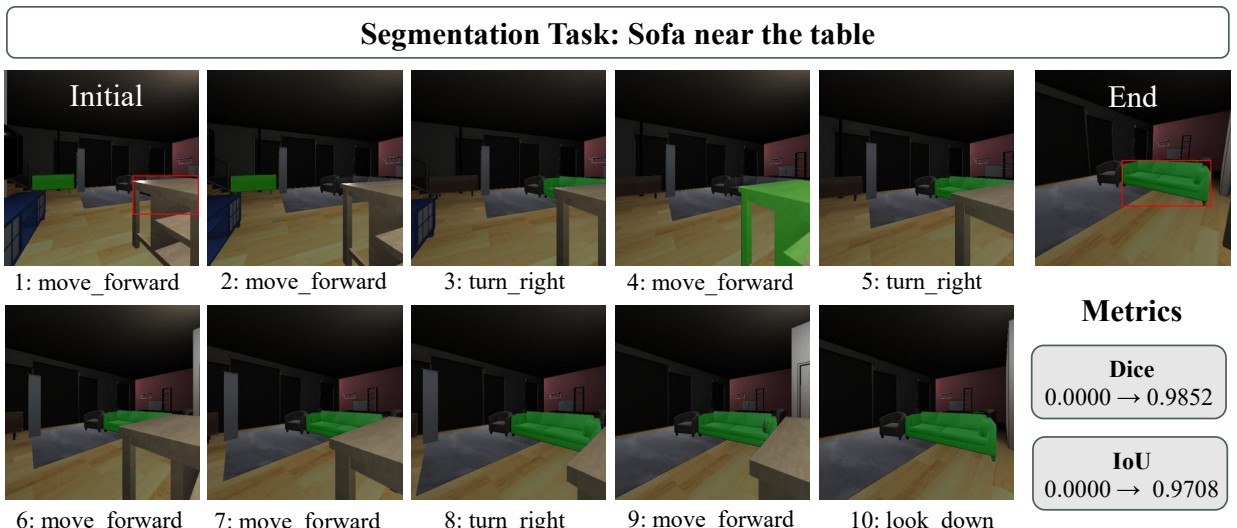

*Figure 9.* **Segmentation example for the task "Sofa near the table".** The initial viewpoint presents severe occlusion and unfavorable framing. After several actions chosen by our VLM policy, the agent acquires a viewpoint with larger visible area and reduced truncation. The SAM mask (overlay) becomes cleaner and better aligned with the object extent, improving both IoU and Dice.

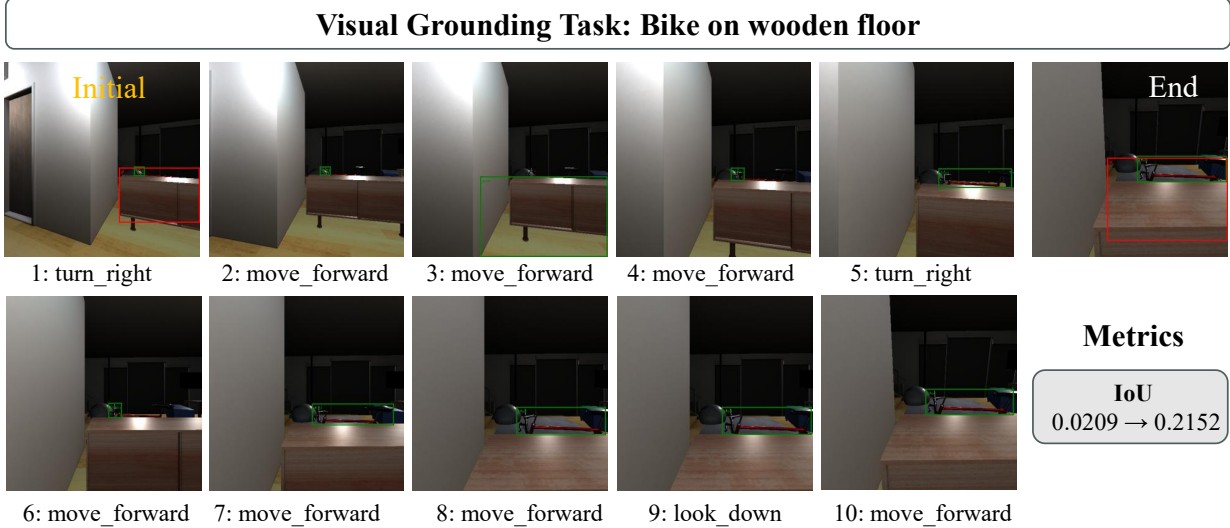

*Figure 10.* **Visual Grounding example for the task "Bike on wooden floor".**

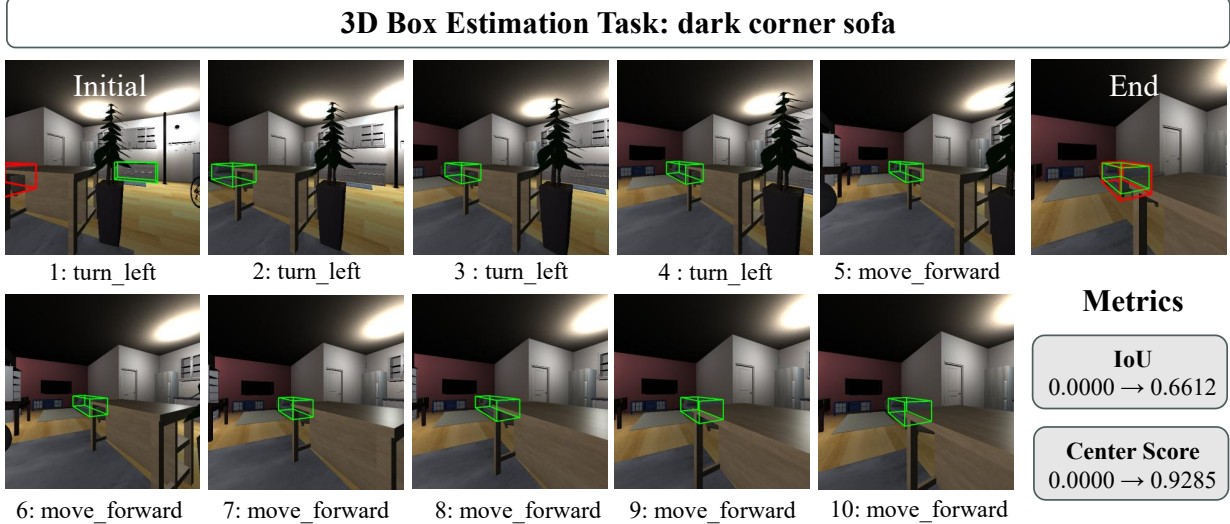

*Figure 11.* **3D Box Estimation example for the task "dark corner sofa".**

*Table 7.* Three perception module baselines on HM3D with different controllers. Visual grounding is evaluated by mAP, segmentation by IoU and Dice, and 3D box estimation by IoU and Center Score (higher is better). The best results are highlighted in **bold**. Degraded results are highlighted in red, while improved results are highlighted in green.

| Perception Module + Policy | Visual Grounding | | | Segmentation | | 3D Box Estimation | |
|---|---|---|---|---|---|---|---|
| | mAP@0.5 ↑ | mAP@0.75 ↑ | mAP$_{avg}$ ↑ | IoU ↑ | Dice ↑ | IoU ↑ | Center Score ↑ |
| Pretrained Perception Module (PPM) | 0.5914 | 0.4224 | 0.5069 | 0.4694 | 0.5460 | 0.3136 | 0.5653 |
| PPM + Forward | 0.3133 (-47.02%) | 0.2475 (-41.40%) | 0.2804 (-44.68%) | 0.2362 (-49.69%) | 0.2700 (-50.55%) | 0.1429 (-54.42%) | 0.2879 (-49.08%) |
| PPM + Random | 0.5032 (-14.91%) | 0.3739 (-11.48%) | 0.4386 (-13.48%) | 0.4052 (-13.68%) | 0.4689 (-14.12%) | 0.2714 (-13.45%) | 0.4866 (-13.92%) |
| PPM + Heuristic | 0.5403 (-8.64%) | 0.4043 (-4.28%) | 0.4723 (-6.83%) | 0.4754 (+1.26%) | 0.5384 (-1.39%) | 0.3343 (+6.60%) | 0.5700 (+0.83%) |
| PPM + Shortest Path | 0.5997 (+1.40%) | 0.4988 (+18.09%) | 0.5493 (+8.35%) | 0.4915 (+4.71%) | 0.5614 (+2.82%) | 0.3316 (+5.74%) | 0.5853 (+3.53%) |
| PPM + InternVL3.5-2B | 0.4512 (-23.71%) | 0.3606 (-14.62%) | 0.4059 (-19.93%) | 0.4745 (+1.09%) | 0.5526 (+1.21%) | 0.3028 (-3.43%) | 0.5482 (-3.03%) |
| PPM + Qwen3VL-2B | 0.4512 (-23.71%) | 0.3606 (-14.62%) | 0.4059 (-19.93%) | 0.4326 (-7.85%) | 0.5007 (-8.29%) | 0.2761 (-11.95%) | 0.4725 (-16.42%) |
| **PPM + Ours (InternVL3.5-2B)** | 0.6400 (+8.21%) | 0.5230 (+23.82%) | 0.5815 (+14.72%) | 0.4981 (+6.12%) | 0.5624 (+3.01%) | 0.3306 (+5.42%) | 0.5748 (+1.67%) |
| **PPM + Ours (Qwen3VL-2B)** | **0.6752** (+14.17%) | **0.5633** (+33.37%) | **0.6193** (+22.16%) | **0.5261** (+12.49%) | **0.5908** (+8.57%) | **0.3420** (+9.31%) | **0.5955** (+5.67%) |

*Table 8.* Experimental proof of catastrophic forgetting under downstream fine-tuning.

| Fine-tuning Epoch | COCO mAP ↑ | Habitat mAP ↑ |
|---|---|---|
| 0 (Zero-shot) | **0.574** | 0.298 |
| 1 | 0.430 | 0.531 |
| 2 | 0.355 | **0.611** |

*Table 9.* Source-domain performance

| Method | COCO mAP ↑ |
|---|---|
| PPM | 0.5031 |
| TTAOD-Foundation | 0.4228 |
| **PPM + Ours** | **0.5031** |

*Table 10.* Robustness to imperfect depth estimation in 3D box estimation.

| Gaussian Noise Std | 3D Box IoU ↑ | Center Score ↑ |
|---|---|---|
| 0.00 | 0.3380 (+27.68%) | 0.6893 (+25.35%) |
| 0.01 | 0.3277 (+23.76%) | 0.6691 (+21.67%) |
| 0.05 | 0.3264 (+23.29%) | 0.6477 (+17.78%) |
| 0.10 | 0.3151 (+19.01%) | 0.6377 (+15.96%) |

*Table 11.* Sensitivity to initial spatial prediction precision.

| Noise Level | Visual Grounding mAP$_{avg}$ ↑ | Segmentation IoU ↑ | 3D Box Estimation IoU ↑ |
|---|---|---|---|
| Light | 0.7873 (+11.02%) | 0.6334 (+12.68%) | 0.3384 (+27.79%) |
| Medium | 0.7517 (+5.99%) | 0.5942 (+5.71%) | 0.3071 (+15.98%) |
| Heavy | 0.7783 (+9.75%) | 0.6346 (+12.90%) | 0.3277 (+23.78%) |

