# OpenReview forum: "See, Act, Adapt: Active Perception for Unsupervised Cross-Domain Visual Adaptation via Personalized VLM-Guided Agent"
_ICML.cc/2026/Conference — ICML 2026 regular_

### Official Review · Reviewer_gjX4 · 2026-02-27

**Soundness:** 3
**Presentation:** 3
**Significance:** 3
**Originality:** 4
**Overall Recommendation:** 5
**Confidence:** 4

**Summary:**

The paper introduces Sea2 (See, Act, Adapt), a framework that addresses the degradation of perception models in new environments due to domain gaps. Rather than adopting the conventional approach of fine-tuning the perception models, which can lead to catastrophic forgetting and requires expensive annotations, the authors propose an active perception paradigm. The research question asks: "Can we adapt perception to new domains without touching the models themselves?" Sea2 addresses this by controlling an agent’s camera pose to seek out more informative viewpoints based on scalar feedback from the frozen perception systems. The framework is trained in two stages. First, a Supervised Fine-Tuning (SFT) stage uses a dataset of trajectories generated by heuristics (object search, proximity adjustment, and viewpoint centering) to align a Vision-Language Model (VLM) with embodied control requirements. Second, the policy is refined via Reinforcement Learning (RL), specifically using Group Relative Policy Optimization, guided by a reward function composed of schema adherence ($r_f$), prediction confidence ($r_c$), and geometric consistency ($r_g$; i.e. spatial consistency between the predicted region and the observation), the agent learns to maximize the cumulative quality of perception outcomes. The proposed method is evaluated on visual grounding, segmentation, and 3D bounding box estimation tasks using the ReplicaCAD, HM3D and photo-realistic Habitat datasets and compared against several baselines.

**Compliance With Llm Reviewing Policy:**

Affirmed.

**Final Justification:**

The authors have addressed all of my questions and concerns in the rebuttal. This paper introduces an effective paradigm to address the timely problem of perception degradation in novel environments without retraining the perception modules, outperforming state-of-the-art baselines across multiple mainstream tasks. By reframing active movement as a form of adaptation, the work offers a well-motivated, original, and rigorously validated contribution to embodied AI.

**Key Questions For Authors:**

- Have the authors evaluated the sensitivity of the RL policy to miscalibrated confidence scores? In scenarios where a perception model is overconfident but wrong, does the agent still converge to informative viewpoints?
- Have you considered using predictive uncertainty or calibrate predictions instead of using raw confidence scores ?
- For tasks involving regression, how sensitive is the geometric consistency check to the precision of the initial bounding box? How do you handle the confidence in such tasks? (as typically confidence is modelled as variance in these cases)

**Limitations:**

no; consider adding a discussion on sensitivity to miscalibated perception modules

**Strengths And Weaknesses:**

*Strengths*
- The paper addresses a critical and highly relevant problem in embodied AI: the loss of perception accuracy in novel environments without resorting to expensive re-training.
- The authors propose an original paradigm by shifting the focus from model adaptation to viewpoint adaptation, essentially treating "acting" as a form of "adapting"
- The method demonstrates substantial performance improvements across multiple mainstream vision tasks (visual grounding, segmentation, and 3D box estimation) compared to state-of-the-art baselines.
- The manuscript is well-written, easy to follow and logically structured, supported by extensive ablations on the reward function design and training strategies.
- Reproducibility is well-supported through the inclusion of an appendix containing detailed training and evaluation protocols.

*Weaknesses*
- The reliance on confidence scores ($r_c$) as a primary reward signal is potentially problematic. Many modern perception models are poorly calibrated, particularly when encountering out-of-distribution (OOD) data in a new domain. If the frozen modules output high-confidence but incorrect predictions, the RL agent may learn to navigate toward viewpoints that reinforce these errors rather than resolving them.
- The geometric reward ($r_g$) relies on consistency between the predicted region and the observation. While this is a clever unsupervised signal, its efficacy depends heavily on the initial quality of the depth or spatial estimation, which may also be compromised in the very domain gaps the paper aims to bridge.

---

> ### Author Rebuttal · Authors · 2026-03-31
>
> We thank the reviewer for the insightful comments. Our responses are below:
>
> **1. Confidently Wrong Prediction [W1, Q1]**
> We agree that perception scores can be miscalibrated in OOD settings. Our reward therefore combines model confidence with geometric consistency terms for target area and center alignment.
>
> To test this directly, we analyze both the **High-Confidence Wrong (HCW)** subset (`initial_conf >= τ`, `initial_iou < 0.5`) and the **Low-Confidence Wrong (LCW)** control group (`initial_conf < τ`, `initial_iou < 0.5`). Full Visual Grounding results are below:
>
> | Setting | τ | Correct Rate (`final_iou >= 0.5`) | Improve Rate (`final_iou > initial_iou`) | Avg ΔIoU |
> |:---:|:---:|:---:|:---:|:---:|
> | VG-HCW | 0.6 | 18/41 (43.90%) | 26/41 (63.41%) | +0.2403 |
> | VG-HCW | 0.7 | 13/26 (50.00%) | 17/26 (65.38%) | +0.2489 |
> | VG-HCW | 0.8 | 8/15 (53.33%) | 9/15 (60.00%) | +0.1833 |
> | VG-LCW | 0.6 | 61/86 (70.93%) | 66/86 (76.74%) | +0.4695 |
> | VG-LCW | 0.7 | 66/101 (65.35%) | 75/101 (74.26%) | +0.4333 |
> | VG-LCW | 0.8 | 71/112 (63.39%) | 83/112 (74.11%) | +0.4239 |
>
> For Segmentation and 3D Box Estimation, we report only the representative `τ = 0.7` summary:
>
> | Setting | Correct Rate (`final_iou >= 0.5`) | Improve Rate (`final_iou > initial_iou`) | Avg ΔIoU |
> |:---:|:---:|:---:|:---:|
> | Seg-HCW | 67/203 (33.00%) | 136/203 (67.00%) | +0.1974 |
> | Seg-LCW | 18/43 (41.86%) | 31/43 (72.09%) | +0.2912 |
> | 3D-HCW | 52/313 (16.61%) | 209/313 (66.77%) | +0.0899 |
> | 3D-LCW | 13/72 (18.06%) | 46/72 (63.89%) | +0.1933 |
>
> Even in HCW cases, the agent improves predictions in 64%--67% of cases across all three tasks. This suggests that the policy does not simply reinforce miscalibrated confidence.
>
> **2. Predictive Uncertainty vs. Raw Confidence [Q2]**
> This is a good suggestion. Our current goal is to validate the core "See, Act, Adapt" paradigm using off-the-shelf frozen perception modules, so we directly use their raw confidence outputs. We agree that predictive uncertainty or calibrated scores could provide a better reward signal, and we will include this direction in Future Work.
>
> **3. Sensitivity to Initial Bounding Box Precision [W2, Q3]**
> To assess how sensitive the geometric consistency check is to the precision of the initial bounding box, we conducted a robustness experiment by injecting varying levels of artificial noise into the spatial predictions (2D boxes, segmentation masks, and 3D boxes) before computing the geometric rewards.
>
> Specifically, the perturbation consists of two components: **center translation noise** (sampled from a Gaussian distribution) and **size scaling noise** (sampled from a Log-Normal distribution). We evaluated three progressive noise levels parameterized by (`std_center`, `std_scale`):
> *   **Light:** (0.02, 0.06)
> *   **Medium:** (0.04, 0.10)
> *   **Heavy:** (0.08, 0.18)
>
> The results below show the performance improvements (compared to the static initial viewpoint) under different noise levels:
>
> | Noise Level | VG mAPavg | Seg IoU | 3D Box IoU |
> |:---:|:---:|:---:|:---:|
> | **Light** | 0.7873 (+11.02%) | 0.6334 (+12.68%) | 0.3384 (+27.79%) |
> | **Medium** | 0.7517 (+5.99%) | 0.5942 (+5.71%) | 0.3071 (+15.98%) |
> | **Heavy** | 0.7783 (+9.75%) | 0.6346 (+12.90%) | 0.3277 (+23.78%) |
>
> Across all three noise levels, the agent consistently remains above the static initial viewpoint baseline. Even with heavy initial spatial noise (e.g., 8% center shift variance and 18% scale variance), it still achieves substantial gains in Visual Grounding (+9.75%), Segmentation (+12.90%), and 3D Box Estimation (+23.78%). This shows that the policy learns to dynamically adjust and refine its spatial understanding over the trajectory, rather than catastrophically failing due to an imprecise initial bounding box.
>
> For tasks involving regression, we currently handle confidence with a simple fixed-weight combination of the raw model confidence and the geometric consistency score. Our goal here is to validate the feasibility of this design: even without explicitly modeling confidence as variance, combining confidence with geometric consistency already provides a stable and useful reward signal, as supported by the robustness results above. We agree that this association can be improved, and future work will explore more principled ways to couple confidence with geometric consistency.

---

> > ### Author Rebuttal · Reviewer_gjX4 · 2026-04-02
> >
> > I thank the reviewers for their thorough response and for addressing my questions.

---

> > > ### Author Response · Authors · 2026-04-07
> > >
> > > We sincerely thank the reviewer for the encouraging feedback and for recognizing our work's strengths.

---

### Official Review · Reviewer_Mzht · 2026-03-12

**Soundness:** 3
**Presentation:** 3
**Significance:** 2
**Originality:** 2
**Overall Recommendation:** 3
**Confidence:** 3

**Summary:**

This paper introduces Sea², an active perception framework designed to improve the performance of pre-trained vision models in novel embodied indoor environments. Rather than fine-tuning the perception models themselves, the authors propose adapting the agent's deployment by learning to navigate to informative viewpoints. The system utilizes a compact Vision-Language Model (VLM) trained as a low-level pose controller via a two-stage pipeline: supervised fine-tuning and unsupervised reinforcement learning (GRPO). Experiments across various vision tasks demonstrate consistent performance gains over passive and heuristic baselines.

**Compliance With Llm Reviewing Policy:**

Affirmed.

**Final Justification:**

I appreciate the authors' efforts to provide additional experiments and to include the TTAOD-Foundation baseline, which improves the quality of the work. Despite these additions, my primary concerns about the evaluation's comprehensiveness remain unresolved, as state-of-the-art continuous TTA methods often incorporate mechanisms, such as regularization, to mitigate catastrophic forgetting. More comprehensive experiments shall be added to demonstrate the superiority of this framework. Thus, I retain my prior score.

**Key Questions For Authors:**

1) How does the imperfect depth estimation affect the final results?

2) What is the performance compared with state-of-the-art TTA baselines for visual modules?

**Limitations:**

Discussions of limitations are missing from this work.

**Strengths And Weaknesses:**

Strengths:
S1: The framework successfully generalizes across diverse, practical perception tasks (visual grounding, segmentation, and 3D box estimation) using off-the-shelf, frozen modules.
S2: By relying entirely on confidence and consistency as reward signals during the RL stage, the method eliminates costly human annotations.
S3: The presentation is easy to follow.

Weakness:
W1: While the paper claims "adaptation", there is no traditional adaptation. In fact, there is essentially no model adaptation, because the paper only adapts the control modules to improve perception quality. As a result, the claim on "adapt" may be somewhat misleading or require clarification.
W2: Since the method strictly prevents weight updates to avoid catastrophic forgetting, the final perception quality is inherently bottlenecked by the capabilities of the frozen model. As a result, the capabilities for adapting to new environments seems limited. Discussions about this limitation are missing.
W3: The 3D box estimate for reward calculation is not perfectly available in the real world. Ablation studies on such estimation errors should be provided.
W4: Importantly, while this paper is about "adaptation", there is no comparison with any TTA baselines, such as TENT and other state-of-the-art baselines, and whether they encounter catastrophic forgetting. For this reason, it is difficult to evaluate the significance of the approach.

---

> ### Author Rebuttal · Authors · 2026-03-31
>
> We thank the reviewer for the constructive and insightful comments. Our responses are below:
>
> **1. Clarification on "Adaptation" [W1]**
> We agree that our method does not perform model-level weight adaptation. The adaptation is instead **system-level**: the agent changes its viewpoint to compensate for frozen-model gaps in novel domains.
> In active perception, adaptation can arise from how a system acquires observations, not only from parameter updates. We will clarify this system-level viewpoint adaptation scope in the Introduction.
>
> **2. Limitation of Frozen Perception Modules [W2]**
> We agree with this point and will add it explicitly to the paper. Keeping the perception modules frozen avoids catastrophic forgetting, but it also caps performance at the pre-trained model's capability. Even with an optimal viewpoint, the system cannot recognize objects outside the model's prior knowledge. Future work will combine active viewpoint selection with parameter-efficient fine-tuning.
>
> **3. Robustness to Imperfect Depth Estimation [W3, Q1]**
> We agree that the depth used for 3D box estimation is imperfect in real-world settings. To evaluate its effect, we conducted an ablation study by injecting Gaussian noise into the depth maps used for the 3D box estimator.
>
> | Gaussian Noise Std | 3D Box IoU | Center Score |
> |:---:|:---:|:---:|
> | 0.00 | 0.3380 (+27.68%) | 0.6893 (+25.35%) |
> | 0.01 | 0.3277 (+23.76%) | 0.6691 (+21.67%) |
> | 0.05 | 0.3264 (+23.29%) | 0.6477 (+17.78%) |
> | 0.10 | 0.3151 (+19.01%) | 0.6377 (+15.96%) |
>
> As the noise standard deviation increases, the performance degrades gracefully but remains highly robust (still providing a +19.01% IoU gain over the baseline at heavy noise), proving the system does not overly rely on perfect depth inputs.
>
> **4. Comparison with TTA Baselines and Catastrophic Forgetting [W4, Q2]**
> Classic Test-Time Adaptation (TTA) methods like TENT are mainly designed for closed-set classification logits and standard entropy minimization, and are therefore not well matched to open-vocabulary detectors such as GroundingDINO. Recent work on vision-language object detector adaptation also notes that standard entropy minimization can amplify confirmation bias and ignore proposal structure in such models (Belal et al., 2025, *VLOD-TTA: Test-Time Adaptation of Vision-Language Object Detectors*). We therefore compare against TTAOD-Foundation (Gao et al., 2025, *Test-Time Adaptive Object Detection with Foundation Model*), a more suitable recent TTA baseline. For a fair comparison, we trained TTAOD-Foundation with a training set of the same scale as the one used for our Visual Grounding policy training, and we also replaced the GroundingDINO version in our pipeline with the default model version supported by the TTAOD-Foundation project. The ReplicaCAD results are below:
>
> | Method | mAP@0.5 | mAP@0.75 | mAPavg |
> |:---:|:---:|:---:|:---:|
> | Pretrained Perception Module (PPM) | 0.8181 | 0.5708 | 0.6945 |
> | TTAOD-Foundation | 0.8389 (+2.54%) | 0.6008 (+5.26%) | 0.7199 (+3.66%) |
> | PPM + Ours | **0.8675 (+6.04%)** | **0.6975 (+22.20%)** | **0.7825 (+12.67%)** |
>
> Our method outperforms TTAOD-Foundation on all three metrics, suggesting that active viewpoint adaptation is more effective than parameter adaptation alone in this setting.
>
> Regarding catastrophic forgetting, we additionally fine-tuned GroundingDINO on 1,800 images collected from the Habitat HM3D target scenes:
>
> | Fine-tuning Epoch | mAP on COCO (Source Domain) | mAP on Habitat (Target Domain) |
> |:---:|:---:|:---:|
> | 0 (Zero-shot) | **0.574** | 0.298 |
> | 1 | 0.430 | 0.531 |
> | 2 | 0.355 | **0.611** |
>
> After only 1 epoch, the target-domain mAP improves, but the source-domain COCO mAP drops sharply from **0.574 to 0.430**. After 2 epochs, it further drops to **0.355**. This directly shows the catastrophic-forgetting trade-off behind weight adaptation and motivates our choice to keep the perception modules frozen.

---

> > ### Author Rebuttal · Reviewer_Mzht · 2026-04-06
> >
> > Thank the authors for their rebuttal, which addresses most of my concerns. However, I do not think it fully resolves my main concern about performance comparisons. The authors provided a single TTA baseline evaluated only on a specific dataset. A more comprehensive comparison against broader TTA methods across datasets is necessary to substantiate the core advantage.
> >
> > Additionally, the catastrophic forgetting experiment may not reflect the state-of-the-art continuous/online test-time adaptation methods, which can mitigate such forgetting. For this reason, I would rather keep my original rating.

---

> > > ### Author Response · Authors · 2026-04-07
> > >
> > > We thank the reviewer for the follow-up comments. We respond to the two concerns separately below.
> > >
> > > **1. Broader TTA Comparisons Across Datasets**
> > > Our current choice of TTA baseline is constrained by the limited availability of reproducible TTA methods for **GroundingDINO**. To the best of our knowledge, **TTAOD-Foundation** is currently the only open-source and reproducible TTA method specifically applicable to this setting that we could reliably evaluate.
> > >
> > > To address the concern about evaluating on only one dataset, we further added a comparison on **HM3D**:
> > >
> > > | Method | mAP@0.5 | mAP@0.75 | mAPavg |
> > > |:---:|:---:|:---:|:---:|
> > > | Pretrained Perception Module (PPM) | 0.6214 | 0.4320 | 0.5267 |
> > > | TTAOD-Foundation | 0.6776 (+9.04%) | 0.4721 (+9.28%) | 0.5749 (+9.14%) |
> > > | PPM + Ours | **0.6799 (+9.41%)** | **0.5605 (+29.75%)** | **0.6202 (+17.75%)** |
> > >
> > > These results are consistent with those on ReplicaCAD: our method remains competitive at mAP@0.5 and shows substantially larger gains at stricter localization metrics and overall mAP, suggesting that active viewpoint adaptation transfers more effectively across datasets than parameter adaptation alone.
> > >
> > > **2. Catastrophic Forgetting Under TTA**
> > > We would like to clarify that the catastrophic-forgetting statement in the paper refers to **fine-tuning the perception model on collected downstream data**, which is what we validated in the first-round rebuttal. We agree that this is not identical to modern continuous/online TTA.
> > >
> > > To address the reviewer’s concern more directly, we further evaluated whether the TTA baseline also suffers degradation on the original source domain. The result on **COCO** is:
> > >
> > > | Method | mAP on COCO |
> > > |:---:|:---:|
> > > | Pretrained Perception Module (PPM) | **0.5031** |
> > > | TTAOD-Foundation | 0.4228 |
> > > | PPM + Ours | **0.5031** |
> > >
> > > This result shows that, although TTAOD-Foundation improves target-domain performance, it also causes a clear drop on the original COCO domain. In contrast, our method preserves source-domain performance because it keeps the perception model frozen and adapts through viewpoint selection instead.

---

### Official Review · Reviewer_9csN · 2026-03-12

**Soundness:** 3
**Presentation:** 3
**Significance:** 3
**Originality:** 3
**Overall Recommendation:** 5
**Confidence:** 3

**Summary:**

This paper introduces Sea2, an active perception pipeline that improves a frozen perception model by training a (navigation) policy tailored to its success. The framework first uses heuristics-based trajectory generation to fine-tune the VLM policy and then uses GRPO to refine the policy with VLM guidance. The system is evaluated on HM3D, MP3D, and ReplicaCAD to demonstrate the effectiveness.

**Compliance With Llm Reviewing Policy:**

Affirmed.

**Final Justification:**

As summarized in my acknowledgement, I will maintain my original score.

**Key Questions For Authors:**

See weaknesses.

**Limitations:**

The authors didn't include a limitation section in the paper; they are encouraged to do so.

**Strengths And Weaknesses:**

Strength:
- The paper is well-written with nice and informative figures.
- The problem is well-motivated, and the method is comprehensive.
- The experimental evaluation is extensive and solid.

Weaknesses:
- The authors mentioned that collecting downstream data and fine-tuning the perception model will have the limitations:"catastrophic forgetting of prior knowledge and the prohibitive cost of acquiring scene-specific annotations". But there are no experimental results supporting this claim.
- The RL part needs to derive step-wise reward from a pre-trained large perception model. How efficient is this? I would like to visualize the learning dynamics of the RL part, and more details for the implementation of this part in L371-377 will also be helpful.

---

> ### Author Rebuttal · Authors · 2026-03-31
>
> We thank the reviewer for the valuable suggestions. Our responses are below:
>
> **1. Experimental Proof of Catastrophic Forgetting [W1]**
> To directly test catastrophic forgetting under downstream fine-tuning, we fine-tuned GroundingDINO on 1,800 images collected from the Habitat HM3D target scenes.
>
> | Fine-tuning Epoch | mAP on COCO (Source Domain) | mAP on Habitat (Target Domain) |
> |:---:|:---:|:---:|
> | 0 (Zero-shot) | **0.574** | 0.298 |
> | 1 | 0.430 | 0.531 |
> | 2 | 0.355 | **0.611** |
>
> After only 1 epoch, the target-domain mAP improves, but the source-domain COCO mAP **drops sharply from 0.574 to 0.430**. After 2 epochs, it further drops to 0.355. This directly supports our motivation: fine-tuning harms prior knowledge, so we keep the perception modules frozen and adapt the viewpoint instead.
>
> **2. RL Efficiency and Learning Dynamics [W2]**
> We agree that the efficiency of step-wise reward computation is important. In our implementation, the pre-trained perception model used to derive the reward takes about **0.1--0.2 s per inference call**, which makes reward computation practical for RL training.
>
> To address the request on learning dynamics, we provide an **anonymous link** with the GRPO training curves: https://github.com/muveowerkljd/ICML-Anonymous/blob/main/README.md. We track four key metrics during RL training: `actor/entropy_loss`, `actor/grad_norm`, `actor/kl_loss`, and `episode/reward/mean`. The curves indicate stable optimization: entropy decreases early and slightly rebounds later, suggesting a transition from rapid policy consolidation to moderate exploration; gradient norm stays within a stable range with only occasional transient spikes and no sustained divergence; KL loss increases gradually, indicating controlled policy updates under KL regularization; and mean episode reward shows a consistent upward trend. Overall, these trends suggest that RL training is stable, properly constrained, and yields clear performance gains.
>
> We clarify the implementation details around L371--377. Our RL codebase is built on **verl**, with **FSDP** and **vLLM** as the training/inference framework. We integrate verl with the **Habitat** simulator to enable agentic RL in indoor environments. The policy is trained with **GRPO** on **2 RTX Pro 6000 GPUs**, and each training run takes about **7 hours**.
>
> **3. Limitations**
> We will add a Limitations section in the final version. Keeping the perception modules frozen avoids catastrophic forgetting, but it also caps performance at the pre-trained model's capability. Even with an optimal viewpoint, the system cannot recognize objects outside the model's prior knowledge. Future work will combine active viewpoint selection with parameter-efficient fine-tuning.

---

> > ### Author Rebuttal · Reviewer_9csN · 2026-04-03
> >
> > Thanks for addressing my concern. I remain positive about this paper.

---

> > > ### Author Response · Authors · 2026-04-07
> > >
> > > We sincerely thank the reviewer for the encouraging feedback and for recognizing our work's strengths.

---

### Official Review · Reviewer_t9tF · 2026-03-13

**Soundness:** 3
**Presentation:** 3
**Significance:** 2
**Originality:** 2
**Overall Recommendation:** 4
**Confidence:** 3

**Summary:**

This paper introduces an active perception framework that adapts to new environments by optimizing camera pose rather than fine-tuning perception modules. A VLM is transformed into a low-level pose controller via a two-stage training paradigm: initially applying supervised fine-tuning on rule-based exploration trajectories, followed by unsupervised RL using reward signals derived exclusively from frozen, off-the-shelf perception modules.

**Compliance With Llm Reviewing Policy:**

Affirmed.

**Final Justification:**

This paper presents a well-motivated approach that rethinks cross-domain visual adaptation as an active viewpoint optimization problem, effectively avoiding the catastrophic forgetting. The authors address my primary concerns, and I am increasing my score to a 4 (Weak Accept).

**Key Questions For Authors:**

Q1. Provide specific details on exactly how the segmentation confidence score is computed.

Q2. How sensitive is the framework to imperfect depth estimation, and how does depth noise quantitatively affect the final perception and trajectory results?

**Limitations:**

The authors should discuss the limitations of the core design.

**Strengths And Weaknesses:**

## Strengths
S1. The paper rethinks domain adaptation as a viewpoint optimization problem, thereby avoiding catastrophic forgetting and annotation costs.

S2. The proposed plug-and-play method effectively accommodates a diverse set of models.

S3. The core idea of optimizing viewpoint rather than adapting the model is well-motivated, and the paper is well-written.

## Weaknesses
W1. The RL stage relies heavily on the confidence scores of the frozen perception modules to calculate the reward. If the perception module is confidently wrong, a very common problem when deploying models in novel, out-of-distribution domains, the agent will be rewarded for finding viewpoints that reinforce incorrect predictions.

W2. High Inference Latency and Computational Overhead: The framework queries a VLM at every step of the trajectory (up to 10 steps) to generate spatial reasoning and select low-level actions. Running a large model introduces severe computational bottlenecks.

W3. The experimental evaluation lacks comparisons against standard TTA baselines.

W4. The 3D bounding box estimator relies on depth data and simple geometric heuristics. The paper does not adequately assess the system's robustness against a noisy depth sensor.

W5. Hyperparameters (such as the $\lambda$ and $\mu$ weighting choices) appear to be fixed without tuning or ablation across different environmental settings.

---

> ### Author Rebuttal · Authors · 2026-03-31
>
> We thank the reviewer for the constructive comments. Our responses are below:
>
> **1. Confidently Wrong Prediction [W1]**
> We agree that "confidently wrong" predictions are challenging. Our reward is therefore not based on confidence alone; it also includes geometric consistency terms for target area and center alignment.
> To test this directly, we analyze both the **High-Confidence Wrong (HCW)** subset (`initial_conf >= τ`, `initial_iou < 0.5`) and the **Low-Confidence Wrong (LCW)** control group (`initial_conf < τ`, `initial_iou < 0.5`). Full Visual Grounding results are below:
>
> | Setting | τ | Correct Rate (`final_iou >= 0.5`) | Improve Rate (`final_iou > initial_iou`) | Avg ΔIoU |
> |:---:|:---:|:---:|:---:|:---:|
> | VG-HCW | 0.6 | 18/41 (43.90%) | 26/41 (63.41%) | +0.2403 |
> | VG-HCW | 0.7 | 13/26 (50.00%) | 17/26 (65.38%) | +0.2489 |
> | VG-HCW | 0.8 | 8/15 (53.33%) | 9/15 (60.00%) | +0.1833 |
> | VG-LCW | 0.6 | 61/86 (70.93%) | 66/86 (76.74%) | +0.4695 |
> | VG-LCW | 0.7 | 66/101 (65.35%) | 75/101 (74.26%) | +0.4333 |
> | VG-LCW | 0.8 | 71/112 (63.39%) | 83/112 (74.11%) | +0.4239 |
>
> For Segmentation and 3D Box Estimation, we report only the representative `τ = 0.7` summary:
>
> | Setting | Correct Rate (`final_iou >= 0.5`) | Improve Rate (`final_iou > initial_iou`) | Avg ΔIoU |
> |:---:|:---:|:---:|:---:|
> | Seg-HCW | 67/203 (33.00%) | 136/203 (67.00%) | +0.1974 |
> | Seg-LCW | 18/43 (41.86%) | 31/43 (72.09%) | +0.2912 |
> | 3D-HCW | 52/313 (16.61%) | 209/313 (66.77%) | +0.0899 |
> | 3D-LCW | 13/72 (18.06%) | 46/72 (63.89%) | +0.1933 |
>
> Even in HCW cases, the agent improves predictions in 64%--67% of cases across all three tasks. This suggests that the policy does not simply reinforce miscalibrated confidence.
>
> **2. High Inference Latency [W2]**
> We agree that VLM queries add latency. This is a general limitation of VLM-based methods such as VLA. Our goal here is to establish feasibility, so we use a lightweight 2B VLM. Future deployments could distill the VLM policy into a smaller controller.
>
> **3. TTA Baselines [W3]**
> We further compared our method with TTAOD-Foundation (Gao et al., 2025, *Test-Time Adaptive Object Detection with Foundation Model*), a recent TTA baseline designed for open-vocabulary detection models. For a fair comparison, we trained TTAOD-Foundation with a training set of the same scale as the one used for our Visual Grounding policy training, and we also replaced the GroundingDINO version in our pipeline with the default model version suppor TTAOD-Foundation project. The results on ReplicaCAD are shown below:
>
> | Method | mAP@0.5 | mAP@0.75 | mAPavg |
> |:---:|:---:|:---:|:---:|
> | Pretrained Perception Module (PPM) | 0.8181 | 0.5708 | 0.6945 |
> | TTAOD-Foundation | 0.8389 (+2.54%) | 0.6008 (+5.26%) | 0.7199 (+3.66%) |
> | PPM + Ours | **0.8675 (+6.04%)** | **0.6975 (+22.20%)** | **0.7825 (+12.67%)** |
>
> Our method consistently outperforms TTAOD-Foundation across all metrics, especially at the stricter mAP@0.75 threshold, showing that active viewpoint adaptation is more effective than parameter adaptation alone in this setting.
>
> **4. Robustness to Depth Noise [W4, Q2]**
> We evaluated our 3D bounding box estimator's robustness by injecting Gaussian noise into the depth maps. As shown below, the performance degrades gracefully but remains highly robust:
>
> | Gaussian Noise Std | 3D Box IoU | Center Score |
> |:---:|:---:|:---:|
> | 0.00 | 0.3380 (+27.68%) | 0.6893 (+25.35%) |
> | 0.01 | 0.3277 (+23.76%) | 0.6691 (+21.67%) |
> | 0.05 | 0.3264 (+23.29%) | 0.6477 (+17.78%) |
> | 0.10 | 0.3151 (+19.01%) | 0.6377 (+15.96%) |
>
> **5. Hyperparameters Tuning [W5]**
> Indeed, we only conducted ablation and tuning of $\lambda$ on ReplicaCAD. Our primary goal is to validate feasibility, and exhaustively searching for the best hyperparameters in every environment is prohibitively expensive. In addition, we have supplemented ablation and tuning experiments for $\mu$ on ReplicaCAD. The results show that $\mu=0.75$ yields the best balance across all three tasks:
>
> | $\mu$ | Grounding (mAPavg) | Segmentation (IoU) | 3D Box (IoU) |
> |:---:|:---:|:---:|:---:|
> | 0.00 | 0.6975 | 0.5515 | 0.2944 |
> | 0.25 | 0.7684 | 0.6033 | 0.2863 |
> | 0.50 | 0.7503 | 0.5841 | 0.3140 |
> | **0.75** | **0.8052** | **0.6516** | **0.3380** |
> | 1.00 | 0.7336 | 0.5725 | 0.2942 |
>
> **6. Segmentation Confidence Score [Q1]**
> The segmentation confidence score is computed as the equally weighted average (0.5 each) of the GroundingDINO bounding box confidence and the SAM mask confidence.
>
> **7. Limitations**
> We will add a Limitations section in the final version. Keeping the perception modules frozen avoids catastrophic forgetting, but it also caps performance at the pre-trained model's capability. Even with an optimal viewpoint, the system cannot recognize objects outside the model's prior knowledge. Future work will combine active viewpoint selection with parameter-efficient fine-tuning.

---

> > ### Author Rebuttal · Reviewer_t9tF · 2026-04-06
> >
> > Thanks to the authors for the rebuttal and the effort in providing additional experiments. I will increase the score to 4.

---

> > > ### Author Response · Authors · 2026-04-07
> > >
> > > We sincerely thank the reviewer for the encouraging feedback and for recognizing our work's strengths.

---

### Decision · Program_Chairs · 2026-04-30

**Decision:**

Accept (regular)

**Comment:**

This paper receives 2x accept, 1x weak accept and 1x weak reject. The reviewers think that the proposed approach is well-motivated in which it rethinks cross-domain visual adaptation as an active viewpoint optimization problem, effectively avoiding the catastrophic forgetting. The paper introduces an effective paradigm to address the timely problem of perception degradation in novel environments without retraining the perception modules, outperforming state-of-the-art baselines across multiple mainstream tasks. The paper is well written, method is comprehensive and the experiments are extensive and solid. Although the negative rating mentioned the need for more experiments, the AC thinks that the strengths in the proposed method and provided experiments are already showing the effectiveness of the performance outweigh the weaknesses. Thus the AC follows the recommendation of the majority reviews to accept the paper.